# SPOP promotes ubiquitination and degradation of MyD88 to suppress the innate immune response

Qinghe Li[ID][☯], Fei Wang[☯], Qiao Wang, Na Zhang, Jumei Zheng, Maiqing Zheng, Ranran Liu, Huanxian Cui, Jie Wen*, Guiping Zhao[ID]*

Institute of Animal Sciences; State Key Laboratory of Animal Nutrition, Chinese Academy of Agricultural Sciences, Beijing, China

☯ These authors contributed equally to this work.
* wenjie@caas.cn (JW); zhaoguiping@caas.cn (GZ)

**Data Availability Statement:** All relevant data are within the manuscript and its Supporting Information files.

## Abstract

As a canonical adaptor for the Toll-like receptor (TLR) family, myeloid differentiation primary response protein 88 (MyD88) has crucial roles in host defense against infection by microbial pathogens, and its dysregulation might induce autoimmune diseases. Here, we demonstrate that the chicken Cullin 3-based ubiquitin ligase adaptor Speckle-type BTB–POZ protein (chSPOP) recognizes the intermediate domain of chicken MyD88 (chMyD88) and degrades it through the proteasome pathway. Knockdown or genetic ablation of *chSPOP* leads to aberrant elevation of chMyD88 protein. Through this interaction, chSPOP negatively regulates NF-κB pathway activity and thus the production of IL-1β upon LPS challenge in chicken macrophages. Furthermore, *Spop*-deficient mice are more susceptible to infection with *Salmonella typhimurium*. Collectively, these findings demonstrate MyD88 as a bona fide substrate of SPOP and uncover a mechanism by which SPOP regulates MyD88 abundance and disease susceptibility.

## Author summary

MyD88 is a central adaptor that mediates initiation of the innate immune response and production of the proinflammatory cytokines that restrain pathogens and activate adaptive immunity. Although MyD88 is crucial for a host to prevent pathogenic infection, misregulation of its abundance might lead to autoimmune diseases. Thus, degradation of MyD88 is a key canonical mechanism for terminating cytokine production. Here, we characterized a novel E3 ligase, SPOP, that targets MyD88 for degradation. ChSPOP attenuated IL-1β production through K48-linked polyubiquitination and degradation of chMyD88, and thus impaired immune responses. *Spop* deficient mice showed more susceptibility to infection by *Salmonella typhimurium*. These findings demonstrate that SPOP is a negative regulator of MyD88-dependent pathway activation triggered by LPS and *Salmonella typhimurium*, which helps the host to maintain immune homeostasis.

**Funding:** This work was supported by grants from National Natural Science Foundation of China (No. 31572393) to Guiping Zhao, the Natural Science Foundation of Beijing (6182032) to Qinghe Li, National Nonprofit Institute Research Grant (2017ywf-zd-5) to Qinghe Li, the Earmarked Fund for Modern Agro-Industry Technology Research System (CARS-41) to Jie Wen, and the Agricultural Science and Technology Innovation Program (ASTIPIAS04) of Chinese Academy of Agricultural Sciences to Jie Wen. The funders had no role in study design, data collection and analysis, decision to publish, or preparation of the manuscript.

**Competing interests:** The authors have declared that no competing interests exist.

## Introduction

The host innate immune system is the first line of defense against invading pathogens, and relies on efficient recognition of microbial agents. Activation of the innate immune response requires the detection of pathogen-associated molecular patterns (PAMPs), including proteins, lipids, carbohydrates, and nucleic acids [1]. These are recognized by pattern recognition receptors such as Toll-like receptors (TLRs), NOD-like receptors, retinoic acid-inducible gene 1 (RIG-I)-like receptors, and C-type lectin receptors [2, 3]. The most widely-used are the Toll-like receptors, which are involved in the recognition of PAMPs like bacterial lipopolysaccharides (LPSs), flagellins, fungi, and viral nucleic acids [4]. Upon recognizing a PAMP, TLRs recruit downstream adaptors such as MyD88 to activate intracellular signaling pathways that result in the production of interferons and proinflammatory cytokines to antagonize the infection [4, 5].

Toll-like receptors are widespread and have been found in both animal and plant phyla, indicating that these receptors are part of an ancient pathogen surveillance system [6]. Most human TLR orthologues have been identified and show functions similar to their counterparts in chickens, such as TLR2–TLR8. As in mammals, the activation of innate immune responses in chickens is crucially affected by TLR-based recognition of pathogens. In chickens, TLR2 recognizes peptidoglycan, TLR4 binds LPS, and TLR5 senses flagellin through mechanisms that are almost the same as in mammals [7]. However, the TLR repertoire of chickens is unique. Chickens lack TLR9, which is responsible for sensing CpG DNA, and instead utilize TLR21 to recognize the unmethylated CpG DNA motifs commonly found in bacteria [8].

As a central adaptor for TLR signaling, MyD88 converts signals from TLRs to activate downstream pathways. MyD88 is subjected to many protein modifications, such as phosphorylation and ubiquitination[9–12]. The protein tyrosine phosphatase Src homology region 2 domain-containing phosphatase-1 (SHP1), encoded by the gene *PTPN6*, suppresses the phosphorylation of MyD88 at tyrosine residues 180 and 278 by spleen tyrosine kinase. Mutation of *PTPN6* has been linked with autoinflammatory and autoimmune diseases, and phosphorylation of MyD88 is a prerequisite for the induction of inflammatory disease in *PTPN6*-mutated mice [9]. Similarly, protein polyubiquitination and deubiquitination have been shown to play critical regulatory roles in host innate immunity. Previous studies have identified E3 ligases that modulate TLR signaling by promoting the polyubiquitination and degradation of MyD88, including Nrdp1, Smurf and Cbl-b [10–12]. MyD88 is also subjected to regulation by protein deubiquitination [13, 14]. The deubiquitinase CYLD negatively regulates TLR–MyD88-dependent signaling by deubiquitinating the nontypeable *Haemophilus influenza*-induced K63-linked polyubiquitination of MyD88 [13]. In addition, phosphorylation of OTUD4 confers K63 deubiquitinase activity, allowing it to deubiquitinate MyD88 and subsequently deactivate TLR-mediated NF-κB signaling [15].

Ubiquitin is a small polypeptide that is covalently added onto proteins by ubiquitin ligase complexes, after which the targeted substrate undergoes proteasome-dependent protein degradation [16]. The substrate specificity of ubiquitin ligation depends on the associated E3 ligase, which recruits substrates through direct protein interaction. The protein SPOP, which selectively binds to substrates via its N-terminal domain, acts as an adaptor for the Cul3-RBX1 E3 ubiquitin ligase complex [17]. SPOP has been linked to the ubiquitination and degradation of a number of proteins in *Drosophila* and humans, including AR, DAXX, SENP7, Ci/Gli, and macroH2A [18–22]. Genome-wide analyses have revealed that *Spop* has a high mutation frequency, in many types of cancer, such as prostate and kidney cancer, with mutations predominantly occur in the substrate-recognizing meprin and TRAF homology (MATH) domain [23].

Previous studies have shown that SPOP plays important roles in tumorigenesis, cell apoptosis, X chromosome inactivation and animal development [19–21, 24]; however, the association between SPOP and host innate immunity remains poorly understood. In this study, we identified SPOP as the ubiquitin ligase adaptor that directly promotes K48-linked polyubiquitylation and destabilizes the MyD88 protein. We also demonstrated that SPOP is critical for regulating NF-κB signaling and innate immune response in *Salmonella* infection.

## Results

### ChSPOP interacts and colocalizes with chMyD88

Since protein ubiquitination has emerged as an important regulatory mechanism for MyD88 signaling, we investigated whether other E3 ubiquitin ligases are involved in the regulation of MyD88. Examining the amino acid sequence of chMyD88, we noticed canonical S/T-rich motifs that correspond to the binding consensus amino acid motif of the SPOP-Cul3-Rbx1 E3 ligase complex [18]. We therefore investigated the association between chMyD88 and chSPOP by constructing expression vectors with the encoding genes and transfecting them into chicken embryonic fibroblasts (DF1) cells. As expected, exogenously introduced chMyD88 interacted with chSPOP *in vitro* (Fig 1A). The same interaction was observed for human and mouse MyD88 and SPOP in human cervical carcinoma cells (Hela cells) and Chinese hamster ovary cells (CHO cells) (S1A and S1B Fig). We also performed immunoprecipitation with an antibody against chSPOP and demonstrated that endogenous chMyD88 could co-immunoprecipitate with chSPOP (Fig 1B). Finally, immunofluorescence analysis consistently demonstrated colocalization of chMyD88 and chSPOP (Fig 1C). Taken together, these data suggest that chSPOP could interact and co-localize with chMyD88.

MyD88 contains an N-terminal death-like (DD) domain, an intermediate (INT) domain, and a C-terminal Toll/interleukin-1 receptor (TIR) homology domain [25]. To further map the elements of chMyD88 that mediate its interaction with chSPOP, we constructed a series of GFP-tagged full-length and truncated chMyD88 mutants and analyzed the interactions between the resulting proteins and Myc-tagged recombinant full-length chSPOP (Fig 1D). We found that chSPOP co-precipitated with wild-type chMyD88, the DD domain truncated mutant, and the TIR domain truncated mutant, but not with the INT domain truncated mutant (Fig 1E), indicating that chMyD88 interacts with chSPOP via its INT domain.

Previous studies have reported that SPOP is comprised of an N-terminal MATH domain, a bric-a-brac, tramtrack, and broad complex (BTB)/POZ domain, and a 3-box domain together with the C-terminal nuclear localization sequence [26]. To identify elements of chSPOP that interact with MyD88, full-length or truncated forms of chSPOP were co-transfected with chMyD88 into DF1 cells. Subsequent co-immunoprecipitation assays showed that only the MATH domain of chSPOP was required for interaction with chMyD88 (Fig 1F and 1G), consistent with the finding that the MATH domain was primarily involved in substrate recognition and binding [17].

### ChSPOP promotes proteasomal degradation of chMyD88

We next examined whether chMyD88 was subject to chSPOP-mediated protein degradation. As expected, exogenously-expressed chSPOP efficiently decreased the expression of chMyD88 in a dose-dependent manner (Fig 2A). However, the mRNA level of *chMyD88* remained unchanged (S2A and S2B Fig), indicating that chSPOP regulates chMyD88 at the translational rather than transcriptional level. Consistent with this finding, knockdown of endogenous *chSpop* in the presence of the translation inhibitor cycloheximide led to an increase in chMyD88 abundance (Fig 2B). The observed decease in chMyD88 due to chSPOP was rescued by the proteasome inhibitors

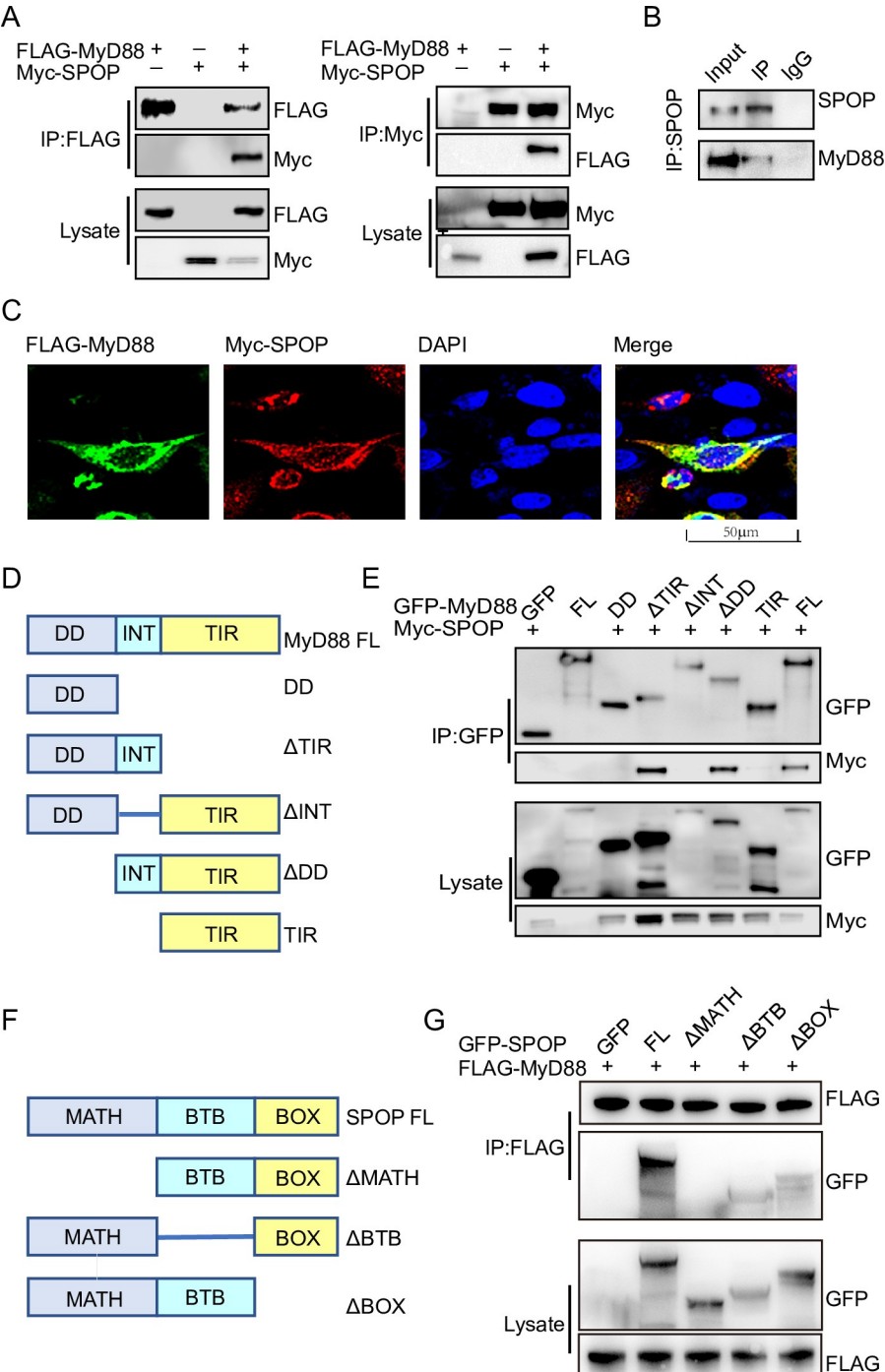

**Fig 1. Interaction of chMyD88 with chSPOP.** (A) Chicken DF1 cells were transfected with indicated the plasmids. Immunoprecipitation was carried out to detect interaction between chMyD88 and chSPOP using the anti-FLAG or anti-Myc antibody, followed by immunoblot analysis with indicated antibodies. (B) Co-immunoprecipitation of endogenous chSPOP and endogenous chMyD88. Cell lysates were immunoprecipitated by anti-SPOP or control IgG antibody, followed by immunoblot with indicated antibodies. (C) Immunofluorescence analysis of chMyD88 and chSPOP. Chicken DF1 cells transfected with FLAG-tagged chMyD88 and Myc-tagged chSPOP. The cells were fixed and incubated with anti-FLAG and anti-Myc antibodies, followed by incubation with secondary antibody. Nuclei were stained with DAPI. The colocalization of chMyD88 and chSPOP was detected by a confocal microscopy. (D) Schematic presentation of chMyD88 and its truncation mutants. FL, full-length; DD, death domain; INT, intermediate domain. TIR, Toll Toll/interleukin-1 receptor homology domain. (E) GFP-tagged chMyD88 or its truncated mutants and Myc-tagged chSPOP were co-transfected into chicken DF1 cells. Cell lysates were immunoprecipitated with anti-

GFP antibody and then immunoblotted with indicted antibodies. (F) Schematic diagram of chSPOP and its truncation mutants. MATH, meprin and TRAF homology domain; BTB, bric-a-brac, tramtrack and broad complex/POZ domain; BOX, 3-box domain together with the C-terminal nuclear localization sequence. (G) GFP-tagged chSPOP or its truncated mutants and FLAG-tagged chMyD88 were co-transfected into chicken DF1 cells. Cell lysates were immunoprecipitated with anti-FLAG antibody and then immunoblotted with indicted antibodies.

MG132 and bortezomib (Figs 2C and S2C) while the autophagy inhibitor bafilomycin A had no effect on chMyD88 degradation (S2C Fig). Together, we conclude that chSPOP promotes the degradation of chMyD88 in a ubiquitin-proteasome-dependent way.

The SPOP protein is highly conserved among different species, with only one amino acid difference between chickens, humans, and mice (S2D Fig). To test conservation of the function of SPOP in downregulating MyD88, we constructed expression vectors with SPOP of human and mouse origin and respectively transfected the plasmid into human lung adenocarcinoma cells (A549 cells) and CHO cells. As expected, SPOP negatively regulates endogenous MyD88 in both human and mouse cells (Fig 2D and 2E), suggesting a highly conserved regulatory role of SPOP on MyD88.

## ChSPOP promotes K48-linked polyubiquitination of chMyD88

Protein ubiquitination is the first step in ubiquitin-proteasome-dependent protein degradation, and SPOP is the substrate recognition adaptor of the SPOP–Cullin 3–RING box 1

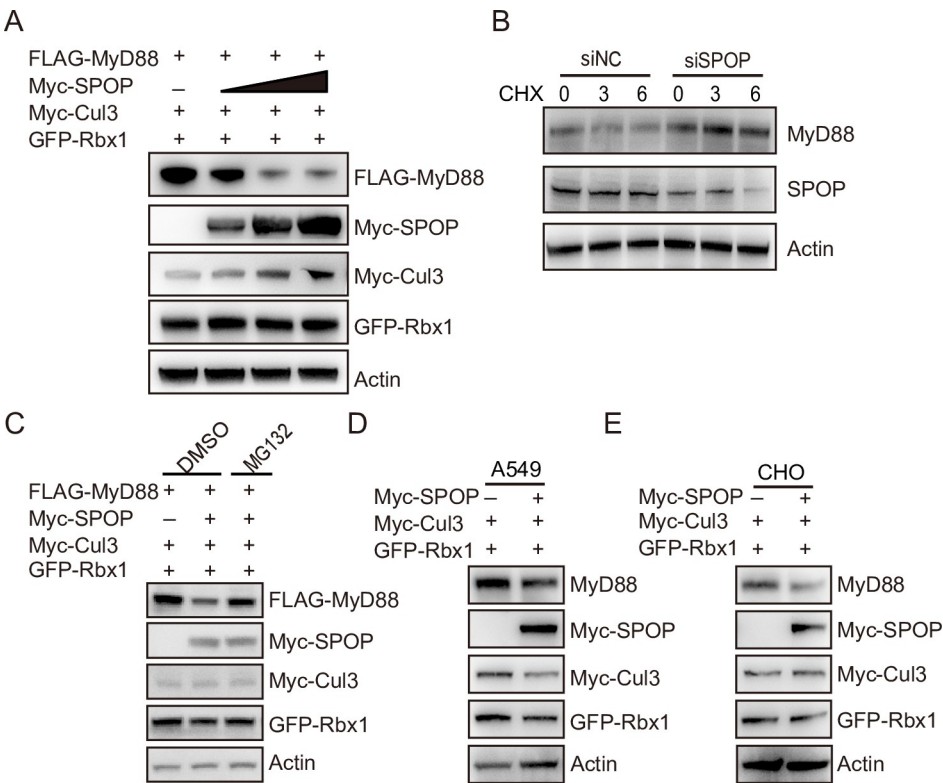

**Fig 2. ChSPOP promotes proteasomal degradation of chMyD88.** (A) Immunoblot analysis of chMyD88 in cell lysates of chicken DF1 cells transfected with chCul3, chRbx1 and increasing doses of Myc-tagged chSPOP (0, 0.4, 0.8, 1.6 μg). (B) Immunoblot analysis of endogenous chMyD88 in cells having *chSpop* inhibited by siRNA. Cells were treated with 50 μg/ml CHX for 0, 3, or 6 hours. (C) Immunoblot analysis of chMyD88 in cell lysates of chicken DF1 cells transfected with chSPOP and treated with DMSO or 20 μM MG132 for 6 hours. (D) and (E) Immunoblot of human MyD88 in A549 cells and mouse MyD88 in mouse CHO cells transfected with Myc-SPOP.

ubiquitin ligase complex. Our above findings uncovered the importance of chSPOP in regulating chMyD88 degradation, so we next questioned whether chMyD88 was an authentic substrate of the chSPOP E3 ligase complex. To explore this possibility, chMyD88 and chSPOP were transfected into chicken cells in the presence of HA-tagged ubiquitin. Our results suggest that in chickens, overexpression of chSPOP increases the ubiquitination level of chMyD88 (Fig 3A, lanes 1 and 2).

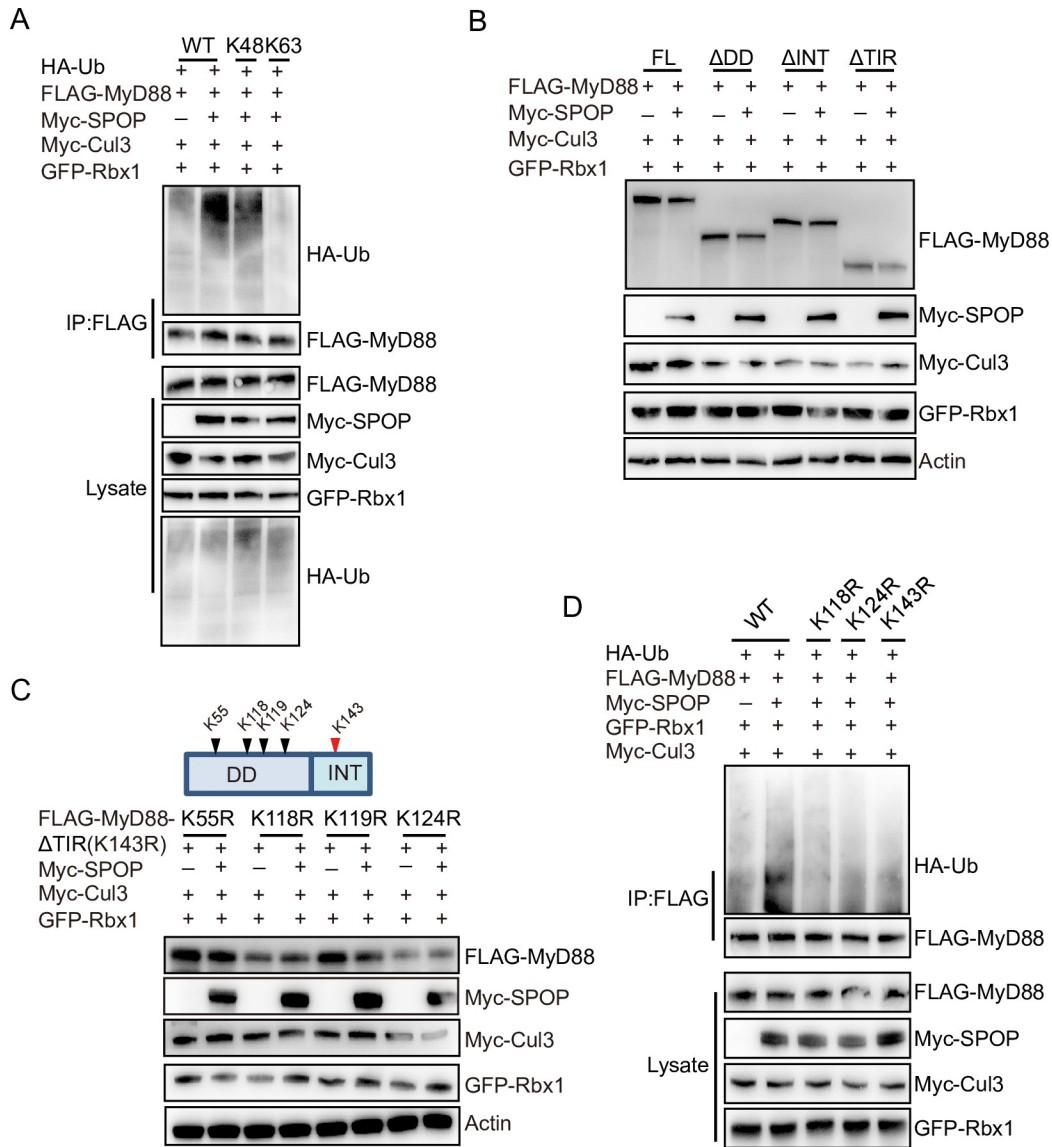

**Fig 3. ChSPOP promotes K48-linked polyubiquitination of chMyD88.** (A) Immunoblot analysis of immunoprecipitated chMyD88 from chicken DF1 cells transfected with indicated plasmids. Immunoprecipitation was carried out with anti-FLAG antibody and then probed with indicated antibodies. (B) Immunoblot analysis of lysates from chicken DF1 cells transfected with indicated plasmids. (C) K118 and K124 of chMyD88 are required for the downregulation of chMyD88 by chSPOP. Truncated chMyD88 K143R constructs consisting of the death-like and intermediate domain with additional K55R, K118R, K119R, or K124R point mutations were transfected into chicken DF1 cells. Expression of the K143R constructs of chMyD88 was detected by anti-FLAG antibody. (D) Mutations at K118, K124, and K143 reduce the polyubiquitination of chMyD88. Wild-type, K118R, K124R, or K143R chMyD88 was transfected into chicken DF1 cells alongside chSPOP. chMyD88 was immunoprecipitated by anti-FLAG antibody, and by immunoblotting performed with indicated antibodies.

MyD88 could be ubiquitinated by either K48- or K63-linked ubiquitination [10–12, 15]. To investigate the molecular mechanism of SPOP-mediated degradation of MyD88, we transfected vectors expressing HA-tagged K48- or K63-linked ubiquitin into chicken DF1 cells and found that overexpression of chSPOP enhanced K48-linked rather than K63-linked ubiquitination of chMyD88 (Fig 3A, lanes 3 and 4). We then substituted the K48 ubiquitin with K48R mutant ubiquitin and found that in the presence of the K48R mutant, SPOP could not promote the ubiquitination of MyD88 (S3A Fig).

We next used truncated forms of chMyD88 to determine the ubiquitination domain of chMyD88. As the INT domain was determined to mediate the interaction between MyD88 and chSPOP, we firstly transfected chicken cells with MyD88 truncations having the DD domain or TIR domain deleted, which both retained the INT domain. Cell lysates were denatured and subjected to immunoblotting to examine the expression of truncated chMyD88, which revealed that chSPOP could downregulate chMyD88 lacking either the DD domain or the TIR domain, but not lacking the INT domain (Fig 3B). We thus co-transfected chSPOP alongside a chMyD88 construct containing only the INT domain, and as expected we observed a significant decrease in INT domain expression (S3B Fig). These results indicate that the intermediate domain alone is sufficient to initiate degradation.

There is only one lysine (K143) in the intermediate domain (S3C Fig), thus we speculated that it might be a ubiquitination site in chMyD88. We subsequently mutated K143 to R143 and investigated the potential of other ubiquitination sites in the DD or TIR domain by co-transfecting chSPOP with mutated full-length or domain deleted chMyD88 into chicken cells. We found that the K143R mutated full-length chMyD88 could still be degraded by chSPOP (S3D Fig, lanes 1 and 2), suggesting there might be other ubiquitination sites besides K143. Furthermore, the K143R mutated TIR domain truncated chMyD88 was susceptible to downregulation by chSPOP, but such activity was abolished when co-transfected with the K143R mutant DD domain truncated chMyD88 (S3D Fig, lanes 3, 4, 5 and 6). These results clearly demonstrate the presence of ubiquitination sites in the DD domain rather than the TIR domain.

The DD domain contains four lysine residues, which we mutated one by one in combination with K143R to determine which mutant would rescue the downregulation of chMyD88 by chSPOP. Immunoblot analysis showed that in combination with K143R, mutation of K118R or K124R, but not K55R and K119R, abolished the degradation (Fig 3C). Furthermore, we generated a triple mutant (K118R, K124R, and K143R) and found that chSPOP completely failed to downregulate the triple-mutant chMyD88 protein (S3E Fig). Lastly, we transfected chSPOP and K118R, K124R or K143R chMyD88 point mutants into chicken DF1 cells and performed immunoprecipitation, which showed that all three mutations reduced the ubiquitination level of chMyD88 (Fig 3D). Taken together, these data suggest that chSPOP promotes the degradation of chMyD88 through ubiquitination on K118, K124, and K143.

## ChSPOP negatively regulates NF-κB activation and proinflammatory cytokine production in chicken macrophages

To assess the effect of chSPOP on proinflammatory responses in chicken macrophages, we overexpressed or knocked down chSPOP in chicken macrophage cells (HD11), treated them with the TLR4 agonist LPS, and measured the expression of IL-1β and IL-8 as indicators of proinflammatory responses. Our results showed that chSPOP overexpression efficiently decreased IL-1β and IL-8 levels in LPS-challenged macrophages (Figs 4A and 4B and S4A). To test the effect of endogenous chSPOP on the host immune response, we silenced its expression using short interfering RNA (siRNA), resulting in a significant decrease of 60% (S4B Fig). Relative to controls, LPS challenge of chSPOP-knockdown macrophages resulted in greater expression of IL-1β and IL-8 (Figs

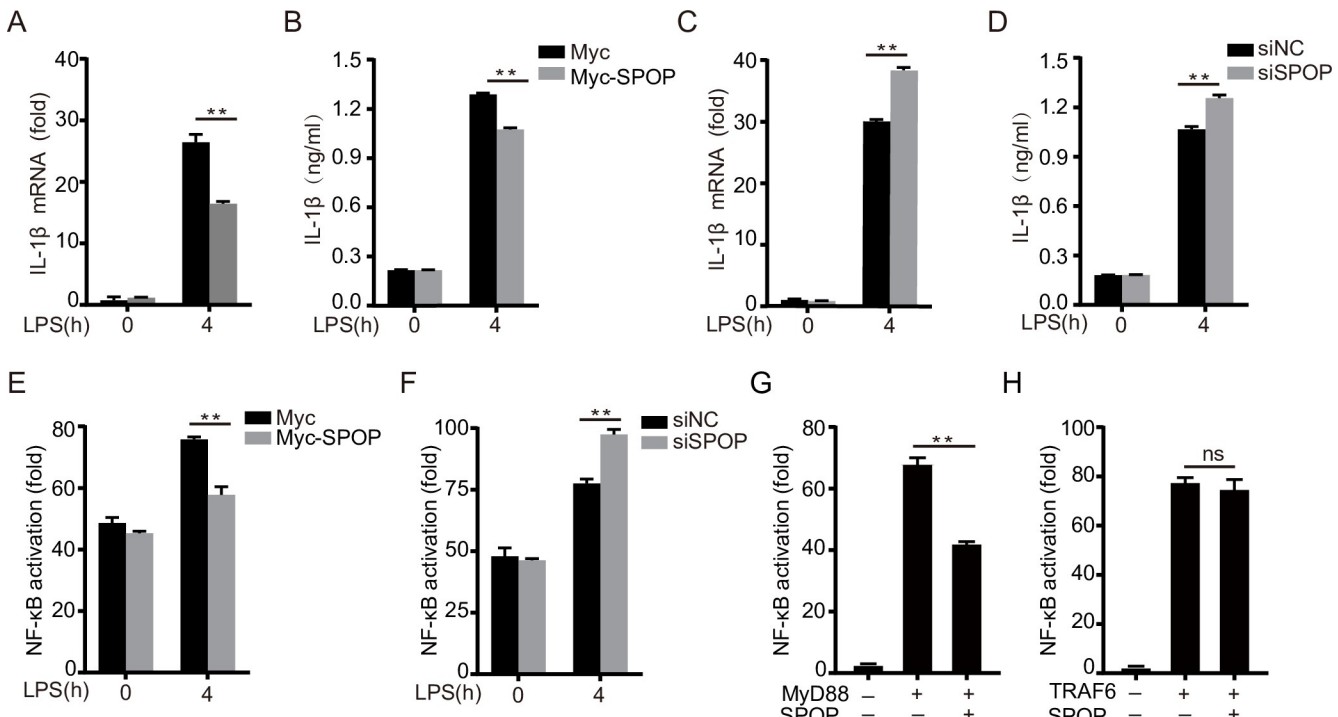

**Fig 4. ChSPOP negatively regulates NF-κB signaling and IL-1β production.** (A) and (B) IL-1β mRNA and protein levels in chicken HD11 macrophages and cell supernatants respectively, with overexpressed chSPOP and stimulated with LPS for 4 h. (C) and (D) IL-1β mRNA level in cells and protein level in cell supernatants from chicken HD11 macrophages transfected with *chSpop* siRNA and stimulated with LPS for 4 h. (E) Relative luciferase activity of NF-κB reporter in chicken HD11 macrophages overexpressing chSPOP. (F) Relative luciferase activity of NF-κB reporter in chicken HD11 macrophages transfected with *chSpop* siRNA. These data are representative of three independent experiments. (G) and (H) Luciferase activity driven by NF-κB promoter in chicken DF1 cells transfected with chSPOP and chMyD88 or chTRAF6. Luciferase assays were performed 24 h after transfection. **$p < 0.01$ and error bars reflect ±SD.

4C and S4C). Moreover, both of impaired chSPOP expression and chMyD88 mutation at ubiquitination sites led to significantly enhanced IL-1β production upon LPS stimulation, as measured by ELISA assay (Figs 4D and S4D). These results support that chSPOP inhibits MyD88 mediated proinflammatory responses in LPS-treated cells.

To identify the molecular mechanisms through which chSPOP inhibits LPS-triggered responses, we performed luciferase assays to evaluate the effect of chSPOP on NF-κB signaling downstream of chMyD88. We expressed exogenous chSPOP or knocked down endogenous *chSpop* by RNAi in chicken macrophage cells transfected with NF-κB reporter, subjected the cells to LPS challenge, and then measured NF-κB activity. As expected, chSPOP negatively regulated LPS-induced NF-κB reporter activation (Fig 4E and 4F). We next assessed whether manipulation of the NF-κB signaling pathway by chSPOP was dependent solely on chMyD88. Through luciferase reporter assays, we determined that chSPOP overexpression inhibited chMyD88-mediated NF-κB activation (Fig 4G), but not NF-κB activation mediated by overexpression of TRAF6, which is downstream of chMyD88 in the pathway (Fig 4H). Taken together, our findings indicate a negative regulatory role of chSPOP on the MyD88-NF-κB signaling pathway and on proinflammatory cytokine secretion.

## SPOP deficiency attenuates host defenses against *Salmonella* infection

To elucidate the *in vivo* function of SPOP, we generated conditional knockout mice using Cre-LoxP recombination (S5 Fig). Germline knockout of *Spop* led to embryonic lethality, and thus

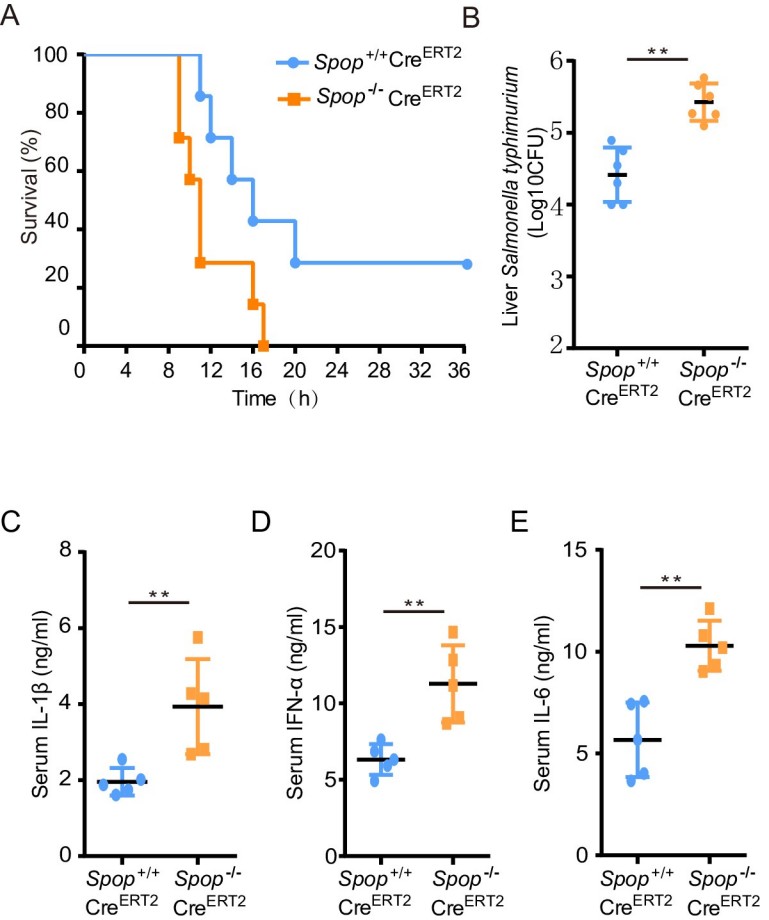

**Fig 5. SPOP deficiency attenuates resistance against *Salmonella* infection in mice.** *Spop* $^{F/F}CreERT2^{+/+}$ knockout mice were generated by Cre-LoxP recombination. Conditional knockout mice were intraperitoneally injected with tamoxifen three times every other day at a dose of 175mg/kg body weight. (A) Mice were challenged with *Salmonella typhimurium* and their survival rates monitored for 36 hours (n = 7 each group; statistical analysis: Mantel–Cox test). (B) Bacterial load in the livers of *Spop* $^{-/-}Cre^{ERT2}$ and *Spop* $^{+/+}Cre^{ERT2}$ mice injected intraperitoneally with *Salmonella typhimurium*. (C), (D), and (E) ELISA for IL-1β, IFN-α and IL-6 in the serum from *Spop* $^{+/+}Cre^{ERT2}$ and *Spop* $^{-/-}Cre^{ERT2}$ mice challenged with *Salmonella typhymurium* for 6 h (n = 5 per genotype). ** $p < 0.01$ and error bars reflect ±SD.

we generated *Spop* conditional knockout mice. *Spop*$^{+/+}$Cre$^{ERT2}$ and *Spop*$^{-/-}$Cre$^{ERT2}$ mice were injected intraperitoneally with *Salmonella typhimurium*, and their survival rates were monitored. *Spop*$^{-/-}$Cre$^{ERT2}$ mice were more susceptible to infection with *Salmonella typhimurium* (Fig 5A), having about ten-fold greater bacteria load in the liver (Fig 5B).

An uncontrolled innate immune response could lead to greater susceptibility to infection [27, 28], therefore we measured proinflammatory cytokines and chemokines in the serum of both groups. The concentrations of IL-1β, IFN-α, and IL-6 were significantly greater in serum from *Spop*$^{-/-}$Cre$^{ERT2}$ mice than in serum from *Spop*$^{+/+}$Cre$^{ERT2}$ controls (Fig 5C–5E). Furthermore, bone marrow derived macrophages (BMDMs) from *Spop*-deficient mice displayed enhanced production of TNF-α, IL-1β, IL-6, and IFN-β (S6A–S6D Fig). In addition to these findings, we observed dramatic MyD88 accumulation and IRAK4 hyper-phosphorylation in the BMDMs from *Spop*$^{-/-}$Cre$^{ERT2}$ mice (S6E Fig).

Guillamot et al. has revealed that hematopoietic-specific deletion of *Spop* in mice leads to lethal neutrophilia after injection with polyinosinic:polycytidylic acid (poly(I:C)) [29]. We

observed similar neutrophilia in *Spop*-deficient mice challenged with *Salmonella typhimurium*. Flow cytometry analysis of peripheral blood from $Spop^{-/-}Cre^{ERT2}$ mice showed a significant expansion of myeloid (CD11b$^+$Gr-1$^+$) cells and decrease of B (B220$^+$) cells (S7A and S7B Fig). Furthermore, the percentages of neutrophils and monocytes were considerably increased, while that of lymphocytes was reduced (S7C Fig).

## Discussion

The recognition of bacterial PAMPs through cellular pattern recognition receptors triggers antibacterial responses to limit bacterial replication. In particular, MyD88-dependent TLRs are key receptors that can detect pathogen-derived LPS, flagellin or single-stranded RNA in the plasma membrane during infection with a variety of bacteria, such as *Salmonella enteritidis* and *Escherichia coli*. MyD88 has been identified as an essential adaptor protein for almost all TLR-dependent signaling [30]. MyD88 is the point of convergence for sensing bacteria and DNA viruses, converting those upstream signals to activate interleukin-1 receptor associated kinases (IRAKs), which ultimately leads to the production of proinflammatory cytokines [31].

Activation of MyD88-NF-κB signaling leads to antimicrobial responses that promote resistance to infection by pathogens. Meanwhile, the half-lives of the inflammation mediators and anti-inflammatory signaling are regulated in order to dynamically modulate inflammatory responses and maintain immune homeostasis. Misregulation of MyD88 signaling results in excessive inflammation and the production of proinflammatory cytokines and interferons that are detrimental to health and might lead to pathological damage, such as cancer and autoimmune diseases [32–34]. Several mechanisms that regulate the activity of MyD88 have been revealed, including ubiquitination, deubiquitination, and phosphorylation [9–13, 15, 20, 29, 35]. A number of E3 ligases have been identified as regulators of MyD88, for example, transforming growth factor-β induces the Smad6-dependent recruitment of E3 ubiquitin ligases Smurf1 and Smurf2 to MyD88, targeting it for proteasomal degradation and thereby displaying its anti-inflammatory function [11]. The E3 ubiquitin ligases Cbl-b and Nrdp1 are also described to polyubiquitinate and degrade MyD88, thereby inhibiting TLR signaling in order to regulate antibacterial or antiviral responses [10, 12]. However, considering that different E3 ligases could be recruited to the same protein substrate, leading to its degradation through multiple mechanisms, more efforts are needed to determine whether MyD88 can be targeted by additional E3 ubiquitin ligases. Here, we uncovered a role for SPOP in the regulation of MyD88 protein abundance and the TLR signaling pathway, and also determined the underlying molecular mechanisms, using chicken as a model organism.

SPOP is an E3 ubiquitin ligase adaptor that is widely expressed in various organs. Emerging evidences suggest that SPOP controls the stability of proteins involved in a range of cellular processes, including tumorigenesis, senescence, transcriptional regulation and apoptosis [17, 18, 20–22]. SPOP has been extensively studied as a tumor suppressor and is a frequently mutated hotspot, most notably in prostate cancer [18, 36, 37]. Cancer-associated SPOP mutants show reduced binding, ubiquitination and degradation of oncoprotein substrates such as the androgen receptor and the ETS transcription factor ERG [18, 38]. In the current study, we characterized the distinct role of SPOP in regulating MyD88 and linked SPOP to innate immune signaling. First, we showed that chSPOP promotes chMyD88 degradation by mediating K48-linked ubiquitination of the chMyD88 residues K118, K124, and K143. As expected, chSPOP recruited chMyD88 to the Cullin 3-SPOP-RBX1 E3 ligase complex through its substrate binding MATH domain. To our knowledge, SPOP is the fourth E3 ligase identified to polyubiquitinate and degrade MyD88, adding to the complexity of the regulation of the MyD88-NF-κB innate immune signaling pathway. Second, we provide further evidence for

the inhibition by chSPOP of chMyD88-induced NF-κB signaling and proinflammatory factor production. The possible effect of SPOP on NF-κB signaling is supported by a recent report showing that downregulation of SPOP promotes the migratory and invasive abilities of osteosarcoma cells through its regulation of the PI3K/Akt/NF-κB signaling pathway. Our findings expand the role of SPOP and uncover its association with innate immune signaling by modulating the adaptor MyD88. Third, we revealed the mechanism by which SPOP regulates MyD88, and showed it to be evolutionarily conserved among birds and mammals. The amino acid sequence of SPOP is almost completely conserved, with only one amino acid substitution among humans, mice, and chickens, while the sequence of MyD88 differs substantially between birds and mammals. However, we confirmed that both human and mouse SPOP interact with and degrade MyD88. Guillamot et al. revealed a critical role for mouse SPOP in restricting inflammation by targeting MyD88 for degradation [29], and a parallel study by Jin et al. has shown that the human SPOP ubiquitin ligase complex suppresses the growth of diffuse large B-cell lymphoma by negatively regulating MyD88/NF-κB signaling [35].

Chickens have been proven to be a versatile experimental model organism in the study of immunology, development biology, virology, and cancer [39]. The current study investigated the downregulation of chMyD88 by chSPOP, and then expanded the investigation into mouse and human cells. We generated a knockout mouse model to illustrate the *in vivo* function of SPOP. Notably, knockout of *Spop* efficiently attenuated resistance to *S. typhimurium* infection in mice. Activated IL-1 signaling was shown to have a critical role in triggering lethal neutrophilia [29, 40]. Consistent with these findings, we observed significant activation of IL-1β production in the serum of *Spop*-deficient mice associated with increased neutrophils infected with *Salmonella typhimurium*, confirming the involvement of IL-1 signaling in *Spop* deletion-induced neutrophilia.

In conclusion, we demonstrate that SPOP-mediated K48-linked ubiquitination and degradation of MyD88 through the proteasome pathway is a novel mechanism that negatively regulates MyD88-dependent proinflammatory signaling.

## Methods

### Ethics statement

Animal care and use protocols were performed in accordance with the regulations in the Guide for the Care and Use of Laboratory Animals issued by the Ministry of Science and Technology of the People's Republic of China. The animal experiments were approved by the Animal Ethics Committee of the Institute of Animal Sciences, Chinese Academy of Agricultural Sciences (Approval Number: IAS2018-8).

### Gene knockout mice, *Salmonella* infection and ethics statement

*Spop* conditional knockout mice were created using classic Cre-LoxP recombination by Beijing Vitalstar Biotechnology Co., Ltd. Exons 4 and 5 of *Spop* were floxed with two *loxp* sites by CRISPR/Cas9-mediated targeting. $Fstl1$-$Cre^{ERT2}$ knock-in mice were obtained from the National Resource Center for Mutant Mice at Nanjing University. Briefly, $Spop^{F/-}$ mice were firstly crossed with $Cre^{ERT2+/+}$ mice to get $Spop^{F/-}Cre^{ERT2+/-}$ heterozygotes, then $Spop^{F/-}Cre^{ERT2}{}^{+/-}$ mice were crossed with individuals of the same genotype to generate $Spop^{F/F}Cre^{ERT2+/+}$ mice. We injected the six-week old mice intraperitoneally with 175 mg/kg bodyweight tamoxifen every other day for 3 times total to drive the expression of Cre. Seven days after the last injection, mice with similar body weight were placed in a pathogen free isolator. $Spop^{-/-}Cre^{ERT2}$ and control mice were then challenged with *Salmonella typhimurium* ($5*10^8$ CFU). Liver samples for bacteria load analysis were collected immediately after the deaths of the mice.

## Cell culture and transfection

Human cervical carcinoma cells (Hela cells, from ATCC), Chinese hamster ovary cells (CHO cells, from ATCC), human lung adenocarcinoma cells (A549 cells, from ATCC), and chicken embryonic fibroblast cells (DF1 cells, from the cell bank of the Chinese Academy of Sciences) were cultured in Dulbecco's modified Eagle's medium supplemented with 10% fetal bovine serum (FBS, Gibco) and 1% penicillin-streptomycin (Gibco). Chicken macrophage cells (HD11 cells, from the cell bank of the Chinese Academy of Sciences) were cultured in RPMI1640 medium (Gibco) complemented with 10% FBS, 5% chicken serum, 1% sodium pyruvate, 1% non-essential amino acids and 1‰ β-mercaptoethanol. These cells were maintained in a humidified incubator with 5% $CO_2$ at 37˚C. Lipofectamine 3000 (Invitrogen) was used for the transfection of plasmids or siRNA into HeLa, CHO and DF1 cells, according to the manufacturer's instructions. TransIT-TKO Transfection Reagent (Mirus Bio) was used for the transfection of siRNA into HD11 cells. For certain experiments, cells were treated with MG132 (20 μM) for 4 h after transfection.

## Antibodies and reagents

Please refer to S1 Table for details.

## Plasmids

*chSpop*, *chMyD88*, *chRbx1 and chCul3* were amplified using standard PCR techniques and high-fidelity DNA polymerase from a spleen cDNA library and were subsequently inserted into the pcDNA3.1 expression vector. Deletion mutants encoding different regions of the chMyD88 or chSPOP proteins were derived from full-length FLAG-chMyD88 or Myc-chSPOP plasmids by PCR and were subcloned into pcDNA3.1. HA-Ub, HA-Ub K48 (all lysines mutated to arginine except for K48), and HA-Ub K63 (all lysines mutated to arginine except for K63) were inserted into the pBI-CMV vector. The Lysine to Arginine point mutants of chMyD88 were generated using the QuikChange mutagenesis kit (Tiangen). All constructs were confirmed by sequencing. The NF-κB-Luc luciferase reporter plasmid was purchased from Promega.

## RNA interference

Duplexed siRNAs were synthesized by Gene-Pharma with the following sequences:

*Spop* siRNA for chicken, 5′- GCCAGAACACUAUGAACAUTT-3′;

Nonspecific siRNA, 5′-UUCUCCGAACGUGUCACGUTT-3′.

## Real-time PCR

Total cellular RNA was extracted by TRIzol (Invitrogen) according to the manufacturer's instructions. Next, cDNA was generated from 1 μg of RNA using the PrimeScript RT reagent kit with gDNA Eraser (Takara). Target mRNAs were quantified by real-time PCR using SYBR Green master mix. Data were normalized to the expression of the housekeeping gene β-actin. The sequences of the PCR primers used to amplify target genes are listed below:

*β-actin*: sense 5′-GAGAAATTGTGCGTGACATCA-3′,

antisense 5′-CCTGAACCTCTCATTGCCA-3′;

*Spop*: sense 5′- AGGCTTGGATGAGGAGAGT -3′,

antisense 5′- CGCTGGCTCTCCATTGCTT -3′;

*MyD88*: sense 5′- TGGAGGAGGACTGCAAGAAGT -3′,

antisense 5′- GCCCATCAGCTCTGAAGTCTT -3′;

*Il-1β*: sense 5′- GCATCAAGGGCTACAAGCTCT -3′,

antisense 5′-T CCAGGCGGTAGAAGATGAAG -3′;

*Il-8*: sense 5′-TCCTCCTGGTTTCAGCTGCT -3′,

antisense 5′- GTGGATGAACTTAGAATGAGTG -3′.

## Immunoprecipation assay and immunoblot analysis

For immunoprecipitation assays, cells transfected with the indicated plasmids were lysed in RIPA buffer (50 mM Tris-HCl, pH 7.4, 150 mM NaCl, 0.25% deoxycholic acid, 1% NP-40 and 0.5% SDS supplemented with protease inhibitor [Roche]) and centrifuged at 12,000 g at 4˚C for 10 min. Whole cell lysate were precleared with protein A/G agarose and then incubated with anti-FLAG beads or appropriate antibody and protein A/G agarose at 4˚C overnight with constant rotation. Immunoprecipitated samples were collected by centrifugation and washed with RIPA buffer three times. After washing, the immunoprecipitates were boiled in sample-loading buffer for 10 min to elute the precipitated proteins, then subjected to immunoblot analysis.

For immunoblot analyses, the protein lysates or immunoprecipitate samples were separated by electrophoresis on SDS-PAGE gels and then transferred onto polyvinylidene fluoride membranes (Millipore). The membranes were first blocked with 5% (w/v) fat-free milk in TBST, then incubated with the corresponding primary antibodies diluted in 5% fat-free milk in TBST. After washing with TBST, the membranes were incubated with the appropriate secondary antibodies diluted in 5% fat-free milk in TBST. The protein bands were visualized using Immobilon Western Chemiluminescent HRP Substrate (Millipore) according to the manufacturer's instructions.

## Immunoflurensence

Chicken DF1 cells transiently transfected with FLAG-MyD88 and Myc-SPOP were cultured for 24h and then fixed by paraformaldehyde. FLAG-MyD88 cells were incubated with an anti-FLAG antibody (Abmart) and Myc-SPOP cells with an anti-Myc antibody (Abmart), both at 1:1000 dilutions. Cells were subsequently incubated with FITC-conjugated secondary antibody (Abcam) for FLAG and Cy3.5-conjugated secondary antibody (Abcam) for Myc. Nuclei were stained with DAPI (Sigma). Finally, cells were finally detected with a confocal microscope (Nikon A1 R MP).

## Luciferase reporter assay

Cells were seeded in 12-well culture plates and transfected with reporter gene plasmids (100 ng) combined with overexpression vector or siRNAs and other constructs as indicated. The pTK-Renilla reporter plasmid was added to normalize transfection efficiency. Twenty-four hours after transfection, LPS or sterile water was added to the cells (to a final concentration of 100 ng/ml). The luciferase activity was determined 4 h later using the Promega luciferase assay kit according to the manufacturer's instructions.

## BMDM isolation

Mouse bone marrow cells were isolated and induced into BMDMS as previously described [41]. At day seven after induction, BMDM cells were treated with 100ng/ml LPS (Sigma) for 6h and cell supernatants collected for ELISA assay.

## Cytokine quantification

Chicken HD11 macrophage cells were transfected with the indicated plasmids or siRNAs for 24 h and then stimulated with 100 ng/mL LPS for 4 h. Afterwards, the culture supernatants were clarified by centrifugation and the level of chicken IL-1β was determined using a chicken IL-1β ELISA kit (LifeSpan BioSciences). Mouse serum cytokines were measured by ELISA kits (Abcam) for mouse IFN-α, IL-1β, and IL-6. For Mouse BMDMs, TNF-α, IFN-β, IL-6 and IL-1β were measured by ELISA using clarified cell supernatants (mouse TNF-α, IFN-β, IL-6, and IL-1β ELISA kits; Abcam).

## Flow cytometry

Whole blood cells were resuspended for 10 min in PBS with 2% FBS. Cells were then incubated with primary antibodies for 30 min at 4˚C. Red blood cells were lysed with ammonium–chloride–potassium buffer. After centrifugation, the remaining cells were resuspended with PBS for flow cytometry analysis. All antibodies were purchased from Biolegend (see S1 Table).

## Statistical analysis

Each experiment was repeated at least three times. Fold changes in mRNA levels (RT-qPCR), reporter assay activity and cytokine content between differently treated samples were compared using one-way ANOVA. In all analyses, $p < 0.05$ was considered statistically significant.

## Supporting information

**S1 Fig. Interaction of human and mouse MyD88 with SPOP.** Hela cells and CHO cells were transfected with human (A) or mouse (B) MyD88 and SPOP. Immunoprecipitation using the anti-FLAG antibody was carried out to detect the interaction, followed by immunoblot analysis with indicated antibodies.
(TIF)

**S2 Fig. Proteasomal degradation of chMyD88 by chSPOP and conservation analysis of SPOP among multiple species.** Expression of *chSpop* and *chMyD88* mRNA in DF1 cells when chSPOP is (A) overexpressed or (B) knocked down. (C) Immunoblot analysis of chMyD88 in cell lysates of chicken DF1 cells transfected with chSPOP and treated with DMSO, 100 nM bafilomycin A, or 50 nM bortezomib. (D) Comparison of SPOP amino acid sequences in human, mouse, and chicken.
(TIF)

**S3 Fig. ChSPOP promotes K48-linked polyubiquitination and degradation of chMyD88.** **(A)** Immunoblot analysis of immunoprecipitated chMyD88 from chicken DF1 cells transfected with indicated expression plasmids. (B) Immunoblot analysis of lysates from chicken DF1 cells transfected with GFP-tagged INT domain of chMyD88 and Myc-chSPOP. (C) Schematic diagram of the truncated chMyD88 mutants. (D) The DD domain of chMyD88 is required for the downregulation of chMyD88 by chSPOP. Indicated expression plasmids were co-transfected into chicken DF1 cells and the cell lysates immunoblotted with corresponding antibodies. (E) ChSPOP failed to downregulate K188, K124, and K143 triple mutated chMyD88 at the protein level. Immunoblot analysis of chMyD88 in cell lysates of chicken DF1 cells co-transfected with Myc-chSPOP.
(TIF)

**S4 Fig. Expression of *Spop* mRNA and pro-inflammatory factors in SPOP overexpressed or inhibited HD11 cell.** (A) Expression of *IL-8* mRNA in chicken HD11 macrophages

overexpressing chSPOP and stimulated with LPS for 4 h. (B) Real-time PCR analysis of *chSPOP* in chicken HD11 macrophage cells transfected with siRNA against *chSPOP*. (C) Expression of *IL-8* mRNA in chicken HD11 macrophages transfected with siRNA against *chSPOP* and stimulated with LPS for 4 h. (D) ELISA of IL-1β in chicken HD11 macrophages transfected with triple mutant MyD88. $^*p < 0.05$, $^{**}p < 0.01$, error bars reflect ±SD.
(TIF)

**S5 Fig. Generation of *Spop* conditional knockout mice.** (A) and (B) Schematic diagram of *Spop* conditional knockout allele. (C) Immunoblot analysis of SPOP in the spleens of $Spop^{+/+}Cre^{ERT2}$ and $Spop^{-/-}Cre^{ERT2}$ mice.
(TIF)

**S6 Fig. BMDMs from $Spop^{-/-}Cre^{ERT2}$ mice are more susceptible to LPS challenge.** (A), (B), (C), and (D) ELISA of TNF-α, IL-1β, IL-6, and IFN-β in BMDMs supernatants from LPS-challenged $Spop^{+/+}Cre^{ERT2}$ and $Spop^{-/-}Cre^{ERT2}$ mice. $^{**}p < 0.01$, error bars reflect ±SD. (E) Immunoblot analysis of BMDMs whole-cell lysates from $Spop^{+/+}Cre^{ERT2}$ and $Spop^{-/-}Cre^{ERT2}$ mice with indicated antibodies.
(TIF)

**S7 Fig. Hematological characteristics of *Spop* conditional knockout mice.** (A) Percentages of B (B220$^+$), T (CD3$^+$), and myeloid (CD11b$^+$Gr-1$^+$) cells in peripheral blood (n = 5). (B) Representative flow cytometry analysis plots of the proportions of B (B220$^+$), T (CD3$^+$), and myeloid (CD11b$^+$Gr-1$^+$) cells in peripheral blood (n = 5). (C) Counts of white and red blood cells and percentages of lymphocytes and neutrophils in peripheral blood of *Salmonella*-challenged $Spop^{-/-}Cre^{ERT2}$ and $Spop^{+/+}Cre^{ERT2}$ mice (n = 5).
(TIF)

**S1 Table. Key resource table for antibodies and reagents used in this study.**
(DOCX)

## Acknowledgments

We thank Z. Du from Institute of Genetics and Developmental Biology, Chinese Academy of Sciences for critical discussions. We thank Beijing Vitalstar Biotechnology Co., Ltd for technical support in generation of conditional knockout mice of SPOP.

## Author Contributions

**Conceptualization:** Qinghe Li.

**Formal analysis:** Qinghe Li, Fei Wang.

**Funding acquisition:** Qinghe Li, Jie Wen, Guiping Zhao.

**Investigation:** Qinghe Li, Fei Wang, Qiao Wang, Na Zhang, Jumei Zheng.

**Methodology:** Qinghe Li, Fei Wang, Maiqing Zheng, Ranran Liu, Huanxian Cui.

**Project administration:** Qinghe Li, Jie Wen, Guiping Zhao.

**Resources:** Qinghe Li, Fei Wang, Guiping Zhao.

**Supervision:** Qinghe Li, Guiping Zhao.

**Validation:** Qinghe Li, Fei Wang.

**Visualization:** Qinghe Li, Fei Wang.

**Writing – original draft:** Qinghe Li, Fei Wang.

**Writing – review & editing:** Qinghe Li, Fei Wang, Jie Wen, Guiping Zhao.

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
