## [Decision Letter · Decision Letter 0]

19 Dec 2019

Dear Dr. Zhao,

Thank you very much for submitting your manuscript "SPOP Promotes Ubiquitination and Degradation of MyD88 to Suppress the Innate Immune Response" (PPATHOGENS-D-19-01994) for review by PLOS Pathogens. Your manuscript was fully evaluated at the editorial level and by independent peer reviewers. The reviewers appreciated the attention to an important problem, but raised some substantial concerns about the manuscript as it currently stands. These issues must be addressed before we would be willing to consider a revised version of your study. We cannot, of course, promise publication at that time. In revising your manuscript, please pay attention to the concerns raised by Reviewer #3, in particular, 

We therefore ask you to modify the manuscript according to the review recommendations before we can consider your manuscript for acceptance. Your revisions should address the specific points made by each reviewer.

(1) A letter containing a detailed list of your responses to the review comments and a description of the changes you have made in the manuscript. Please note while forming your response, if your article is accepted, you may have the opportunity to make the peer review history publicly available. The record will include editor decision letters (with reviews) and your responses to reviewer comments. If eligible, we will contact you to opt in or out.

(2) Two versions of the manuscript: one with either highlights or tracked changes denoting where the text has been changed; the other a clean version (uploaded as the manuscript file).

Additionally, to enhance the reproducibility of your results, PLOS recommends that you deposit your laboratory protocols in protocols.io, where a protocol can be assigned its own identifier (DOI) such that it can be cited independently in the future. For instructions see http://journals.plos.org/plospathogens/s/submission-guidelines#loc-materials-and-methods

We hope to receive your revised manuscript within 60 days. If you anticipate any delay in its return, we ask that you let us know the expected resubmission date by replying to this email. Revised manuscripts received beyond 60 days may require evaluation and peer review similar to that applied to newly submitted manuscripts.

[LINK]

Sincerely,

Zhao-Qing Luo

Associate Editor

PLOS Pathogens

Nina Salama

Section Editor

PLOS Pathogens

Kasturi Haldar

Editor-in-Chief

PLOS Pathogens

orcid.org/0000-0001-5065-158X

Grant McFadden

Editor-in-Chief

PLOS Pathogens

orcid.org/0000-0002-2556-3526

Reviewer's Responses to Questions

**Part I - Summary**

Reviewer #1: In the manuscript titled as “SPOP Promotes Ubiquitination and Degradation of MyD88 to Suppress the Innate Immune Response” by Dr. Zhao and the colleagues. Starting from the consensus sequences for SPOP-interacting domain, they went on to demonstrate that SPOP as an E3 ubiquitin ligase adaptor interacted with, and ubiquitylated MyD88, a key regulator in innate immune response, thus targeting MyD88 for degradation. On functional side, they clearly demonstrated that SPOP-mediated MyD88 ubiquitylation led to marked suppression of LPS-activated NF-kb signaling. Having mapped the sites for ubiquitylation in MyD88, they have nicely shown that ubiquitylation on some of the sites were specifically required for suppressing the activation of NF-kb pathway by LPS. Using siRNA and SPOP KO mice, the authors were able to show that endogenous SPOP was indeed a negative regulator of innate immune response, and SPOP deficient mice were susceptible to microbe infection, in this case, Salmonella typhimurium.

Overall, this work was well thought of and executed in a technically sound way. This work should be of intrest by identifying SPOP as a new regulator of innate immune response.

Reviewer #2: In this paper, the authors identify an interaction between SPOP and MyD88 and attempt to demonstrate that SPOP promotes the degradation of MyD88 through K48-linked ubiquitination, and thus regulates its function. I understand that two other groups published on this very topic recently (Jin et al, Leukemia 2019; Guillamot et al, Nat Immunol 2019) and therefore these authors are eager to publish their work. Overall I am convinced that SPOP is in fact a regulator of MyD88 and am supportive of publication, but some major issues need to be addressed first.

Reviewer #3: This manuscript from Quinghe Li. et al., claims to describe a new role for the E3 ubiquitin ligase adaptor SPOP in controlling innate immune signaling by promoting MYD88 ubiquitination and subsequent degradation. However, two other papers have recently described very similar results and conclusions (Guillamot et al., Nature Immunology 2019 and Jin X et al., Leukemia 2019). Hence, it is very surprising that the authors are not aware of these publications and they do not make any reference to these studies. In addition to the lack of novelty, the conditional SPOP KO mouse model is poorly described, the in vivo experiments are not properly designed and therefore, the results are inconclusive. Taken together all these issues, I recommend that the authors strength their work before considering it for publication.

**Part II – Major Issues: Key Experiments Required for Acceptance**

Reviewer #1: However, when coming to the detailed dissection of the mechanism, several key experiments were missing that will be listed below, not necessarily in the order of relative importance:

1. When address the effect of POP-mediated MyD88 ubiquitylation on the stability of MyD88, a chase experiment using cycloheximide (CHX) should be performed to exclude the potential interference at the level of MyD88 transcription and/or its mRNA translation. Inhibitors like bortezomib and bafilomycin A specific for proteasome and autophagy pathway, respectively, should be tested on MyD88 stability in presence or absence of SPOP.

2. In the current version of the manuscript, it was shown that POP-mediated MyD88 ubiquitylation mainly resulted in poly-Ub chains in K48 linkages (Fig. 3A). Typically, this should be complemented with the experiment using K48R Ub mutant.

Reviewer #2: Figure 2B, 2D and 2E are not very convincing. For 2B: Perhaps the siRNA experiment should be repeated for improved knockdown of SPOP. For 2D-2E: It may help to use a commercially available antibody for endogenous MyD88 to see a greater effect on the overexpression of SPOP.

Much of Figure 3 is not convincing. Specifically, for the deletion analysis in Figure 3B-3C, this needs to be done using the same tag (ie Flag) on the same gel, with proper controls (i.e., WT Flag-MyD88). Also, what happens with Delta-INT in these experiments? I suggest Figure 3E simply be removed as it is not clear what is happening with these combination mutations, and to go straight into the point mutation analysis of Figure 3F.

Reviewer #3: Major points:

1. First of all, the authors should cite the previous work and clarify: 1) which experiments are replicates/similar to already publish work and which are novel findings, 2) how their results correlate with already published data.

2. Figure 3 describes the different lysines in chMYD88 which are ubiquitinated by SPOP. On this figure relies the novelty of this study as many of the other experiments/conclusions had already been shown. Therefore, I would recommend to complement and strength this data by corroborating similar observations in human MYD88. In addition, does over-expression MYD88 mutants induced increased pro-inflammatory cytokine release?

3. Figure 5 describes an in vivo function of SPOP in response to infections. However, the authors do not demonstrate that deletion of SPOP is efficient in their conditional model the and experiments are not properly design and the conclusions are unclear. Specifically:

a. The authors claimed that they generated a conditional KO mouse but they do not provide any description on how they made it. Which exons are targeted? Which specific Cre are they using?

b. In the methods the authors explain that they generate “Spop-/- mice by crossing Spop-/- with Cre transgene C57BL/6.” This is wrong and confusing. You couldn’t generate any conditional KO mice following this statement.

c. In the legend of Figure 5 the authors state: “Spop-/- heterozygous”. Again this is confusing as “-/-“indicates homozygosis

d. In the legend of Figure 5 the authors indicate that Spop was deleted by “feeding the mice with tamoxifen three times every 48 hours”. This is not the proper protocol to delete a gene with a Tamoxifen food and it raises concerns on the efficiency of the deletion.

e. They authors should provide western blot demonstrating that SPOP is efficiently deleted

f. To demonstrate that the phenotype is hematopoietic intrinsic, the authors should generate hematopoietic chimeras

g. The infection is done in only 4 mice and the differences in the survival curve relies in only one control. Therefore, authors should include more mice to ensure that the results are not biased by one outlier.

h. If deletion of SPOP results in MYD88 accumulation and subsequently, increased NF-kB response and pro-inflammatory cytokines release, one would expect an enhanced inflammatory response. Hence, it is unclear why the mice are more susceptible to infection rather than being able to clear more efficiently the pathogen.

i. Are pro-inflammatory cytokines increased in the serum of Spop KO mice? This is important because a sepsis shock could also explain the increase mortality of the mice.

j. Are the percentages of monocytes or neutrophils higher in the peripheral blood of the Spop KO mice than in the wild-type controls?

k. If the mice are more susceptible to infections, are Spop macrophages dysfunctional? How does it correlate with Myd88 dependent pathway?

**Part III – Minor Issues: Editorial and Data Presentation Modifications**

Reviewer #1: Minor issues:

Labels in many panels need to be more carefully tailored to help readers better understand the results. For example, in Fig.1D, for SPOP, the three major domains should be presented in FL, and aligned labels (those used in Fig.1E) of the full-length SPOP should be provided.

Language may need to be polished in many parts of the manuscript for accuracy and fluence.

Reviewer #2: Figure 4D: Is it possible that an insufficient knockdown is giving this very modest result? Can the authors repeat it and try to improve the knockdown efficiency?

Figure 5A: Not many mice were used in the two arms of the experiment (four each). What is the statistical analysis for this result (4/4 dying versus 3/4 dying)?

Abstract: Make it clear that these studies were done primarily in chicken (i.e., it is unclear to the reader what ‘chMyD88’ and ‘chSPOP’ are until later in the manuscript).

Introduction: Many other deubiquitinases besides OTUD4 regulate MyD88, and these should be discussed

The immunofluorescence analysis in Figure 1C is not useful and only reveals that the proteins are both present in the cytosol (even this is rather unclear). Either much higher resolution is necessary, or some other method should be used to demonstrate interaction (e.g., PLA), or this panel should just be removed.

It would be useful to label the different deletion mutants on the side of the panels with the names used in the adjacent panels in Figure 1 (i.e., “Delta-TIR” next to the “DD+INT”)

Some of the language used in the manuscript is slightly confusing and perhaps the manuscript could be improved by careful editing for language. For example, “To test whether the downregulation of SPOP on MyD88 is a common event among mammals” I believe the authors mean “To test whether the function of SPOP in downregulating MyD88 is conserved.”

Reviewer #3: 1. Authors should provide more information about the antibodies used in their experiments.

2. Do bone marrow macrophages from Spop KO mice show increased pro-inflammatory cytokines (TNF-a, il-1b, Il-6…) levels following LPS stimulation?

3. Quality of Figure 1C should be improved.

4. Figure 4C are very small and authors should include more numbers to ensure reproducibility.

PLOS authors have the option to publish the peer review history of their article (what does this mean?). If published, this will include your full peer review and any attached files.

Reviewer #1: No

Reviewer #2: No

Reviewer #3: No

---

## [Editor Report · Decision Letter 1]

15 Apr 2020

Dear Dr. Zhao,

We are pleased to inform you that your manuscript 'SPOP Promotes Ubiquitination and Degradation of MyD88 to Suppress the Innate Immune Response' has been provisionally accepted for publication in PLOS Pathogens.

Best regards,

Zhao-Qing Luo

Associate Editor

PLOS Pathogens

Nina Salama

Section Editor

PLOS Pathogens

Kasturi Haldar

Editor-in-Chief

PLOS Pathogens

orcid.org/0000-0001-5065-158X

Michael Malim

Editor-in-Chief

PLOS Pathogens

orcid.org/0000-0002-7699-2064
---

## [Editor Report · Acceptance letter]

24 Apr 2020

Dear Dr. Zhao,

We are delighted to inform you that your manuscript, "SPOP Promotes Ubiquitination and Degradation of MyD88 to Suppress the Innate Immune Response," has been formally accepted for publication in PLOS Pathogens.

Best regards,

Kasturi Haldar

Editor-in-Chief

PLOS Pathogens

orcid.org/0000-0001-5065-158X

Michael Malim

Editor-in-Chief

PLOS Pathogens

orcid.org/0000-0002-7699-2064